# Activation of the Unfolded Protein Response (UPR) Is Associated with Cholangiocellular Injury, Fibrosis and Carcinogenesis in an Experimental Model of Fibropolycystic Liver Disease

**DOI:** 10.3390/cancers14010078

**Published:** 2021-12-24

**Authors:** Chaobo Chen, Hanghang Wu, Hui Ye, Agustín Tortajada, Sandra Rodríguez-Perales, Raúl Torres-Ruiz, August Vidal, Maria Isabel Peligros, Johanna Reissing, Tony Bruns, Mohamed Ramadan Mohamed, Kang Zheng, Amaia Lujambio, Maria J. Iraburu, Leticia Colyn, Maria Ujue Latasa, María Arechederra, Maite G. Fernández-Barrena, Carmen Berasain, Javier Vaquero, Rafael Bañares, Leonard J. Nelson, Christian Trautwein, Roger J. Davis, Eduardo Martinez-Naves, Yulia A. Nevzorova, Alberto Villanueva, Matias A. Avila, Francisco Javier Cubero

**Affiliations:** 1Department of Immunology, Ophthalmology and ENT, Complutense University School of Medicine, 28040 Madrid, Spain; bobo19820106@gmail.com (C.C.); travelerhhang@gmail.com (H.W.); liuchennj@hotmail.com (H.Y.); agustito@ucm.es (A.T.); zhengkang2016@gmail.com (K.Z.); emnaves@med.ucm.es (E.M.-N.); yulianev@ucm.es (Y.A.N.); 212 de Octubre Health Research Institute (imas12), 28041 Madrid, Spain; 3Department of General Surgery, Wuxi Branch of Zhongda Hospital, Southeast University, Wuxi 214105, China; 4Department of Hepatic-Biliary-Pancreatic Surgery, The Affiliated Drum Tower Hospital of Nanjing University Medical School, Nanjing 210008, China; 5Department of Anesthesia, Zhongda Hospital, Southeast University, Nanjing 210008, China; 6Molecular Cytogenetics and Genome Editing Unit, Human Cancer Genetics Program, Centro Nacional de Investigaciones Oncológicas (CNIO), 28029 Madrid, Spain; srodriguezp@cnio.es (S.R.-P.); truizraul27@gmail.com (R.T.-R.); 7Laboratorio de Investigación Traslacional (LRT1)-ProCURE, Institut Català d’Oncologia (ICO)-IDIBELL, L’Hospitalet de Llobregat, 08908 Barcelona, Spain; avidal@bellvitgehospital.cat (A.V.); avillanueva@iconcologia.net (A.V.); 8Institute of Pathology, Hospital Universitario Gregorio Marañon, 28007 Madrid, Spain; isabel.peligros@salud.madrid.org; 9Department of Internal Medicine III, University Hospital RWTH, 52074 Aachen, Germany; jreissing@ukaachen.de (J.R.); tbruns@ukaachen.de (T.B.); mmohamed@ukaachen.de (M.R.M.); ctrautwein@ukaachen.de (C.T.); 10Liver Cancer Program, Division of Liver Diseases, Department of Medicine, Tisch Cancer Institute, Icahn School of Medicine at Mount Sinai, New York, NY 10029, USA; amaia.lujambio@mssm.edu; 11The Precision Immunology Institute, Icahn School of Medicine at Mount Sinai, New York, NY 10029, USA; 12Graduate School of Biomedical Sciences at Icahn School of Medicine at Mount Sinai, New York, NY 10029, USA; 13Department of Oncological Sciences, Icahn School of Medicine at Mount Sinai, New York, NY 10029, USA; 14Department of Biochemistry and Genetics, University of Navarra, 31008 Pamplona, Spain; miraburu@unav.es; 15Hepatology Program, Cima, University of Navarra, 31008 Pamplona, Spain; lcolyn@unav.es (L.C.); mulatasa@unav.es (M.U.L.); macalderon@unav.es (M.A.); magarfer@unav.es (M.G.F.-B.); cberasain@unav.es (C.B.); 16Centro de Investigación Biomédica en Red de Enfermedades Hepáticas y Digestivas (CIBEREHD), Instituto de Salud Carlos III, 28029 Madrid, Spain; javiervaq@hotmail.com (J.V.); rbanares@ucm.es (R.B.); 17Instituto de Investigaciones Sanitarias de Navarra-IdiSNA, 31008 Pamplona, Spain; 18Instituto de Investigación Sanitaria Gregorio Marañón (IiSGM), 28007 Madrid, Spain; 19Servicio de Aparato Digestivo, Hospital General Universitario Gregorio Marañón, 28007 Madrid, Spain; 20Institute for Bioengineering (IBioE), Faraday Building, The University of Edinburgh, Edinburgh EH8 9AB, UK; L.Nelson@ed.ac.uk; 21Howard Hughes Medical Institute, University of Massachusetts Medical School, Worcester, MA 01655, USA; roger.davis@umassmed.edu

**Keywords:** c-Jun N-terminal kinases (JNK), fibropolycystic liver disease, cholangiocarcinoma (CCA), endoplasmic reticulum (ER) stress, thioacetamide (TAA), CM272

## Abstract

**Simple Summary:**

Polycystic liver disease (PLD) is a group of rare disorders that result from structural changes in the biliary tree development in the liver. In the present work, we studied alterations in molecular mechanisms and signaling pathways that might be responsible for these pathologies. We found that activation of the unfolded protein response, a process that occurs in response to an accumulation of unfolded or misfolded proteins in the lumen of the endoplasmic reticulum, as well as the scarring of the liver tissue, contribute to the pathogenesis of PLD and the development of cancer. As a preclinical animal model we have used mutant mice of a specific signaling pathway, the c-Jun N-terminal kinase 1/2 (*Jnk1/2*). These mice resemble a perfect model for the study of PLD and early cancer development.

**Abstract:**

Fibropolycystic liver disease is characterized by hyperproliferation of the biliary epithelium and the formation of multiple dilated cysts, a process associated with unfolded protein response (UPR). In the present study, we aimed to understand the mechanisms of cyst formation and UPR activation in hepatocytic c-Jun N-terminal kinase 1/2 (*Jnk1/2*) knockout mice. Floxed JNK1/2 (*Jnk^f/f^*) and *Jnk^∆hepa^* animals were sacrificed at different time points during progression of liver disease. Histological examination of specimens evidenced the presence of collagen fiber deposition, increased α-smooth muscle actin (αSMA), infiltration of CD45, CD11b and F4/80 cells and proinflammatory cytokines (*Tnf*, *Tgfβ1*) and liver injury (e.g., ALT, apoptosis and Ki67-positive cells) in *Jnk^∆hepa^* compared with *Jnk^f/f^* livers from 32 weeks of age. This was associated with activation of effectors of the UPR, including BiP/GRP78, CHOP and spliced XBP1. Tunicamycin (TM) challenge strongly induced ER stress and fibrosis in *Jnk^∆hepa^* animals compared with *Jnk^f/f^* littermates. Finally, thioacetamide (TAA) administration to *Jnk^∆hepa^* mice induced UPR activation, peribiliary fibrosis, liver injury and markers of biliary proliferation and cholangiocarcinoma (CCA). Orthoallografts of DEN/CCl_4_-treated *Jnk^∆hepa^* liver tissue triggered malignant CCA. Altogether, these results suggest that activation of the UPR in conjunction with fibrogenesis might trigger hepatic cystogenesis and early stages of CCA.

## 1. Introduction

JNKs are a family of evolutionarily conserved mitogen-activated protein kinases (MAPKs) activated by tumor necrosis factor (TNF) that play an important role in converting extracellular stimuli into a wide range of cellular responses, including inflammatory response, stress response, cell death, cell differentiation and cell proliferation [1,2]. JNKs are encoded by three genes, *Jnk1*, *Jnk2* and *Jnk3*; the products of two of these genes, JNK1 and JNK2, are expressed in the liver [3]. Importantly, JNK1 and JNK2 elicit redundant but also distinct functions [4,5]. In order to characterize the combined functions of the JNK genes, using cell-type-specific deletion models is essential. By implementing this strategy, we were the first to report that aged hepatocyte-specific *Jnk1/2* (*Jnk^∆hepa^*) knockout mice present bile duct hyperplasia. Moreover, the deletion of *Jnk1/2* in an experimental model of chronic liver disease was strongly associated with cell death, cholestasis and cholangiocyte proliferation [1]. A few months later, Manieri and colleagues [6] reported that changes in bile acid metabolism in *Jnk^∆hepa^* mice may contribute to cholangiocyte proliferation and hepatoblast maturation, causing bile duct hyperplasia and cholangiocyte injury, which leads to cholangiocarcinoma (CCA) development at late stages. A very recent paper [7] confirmed these findings and identified a molecular link between JNK and RIPK1 as a mechanistic trigger of murine polycystic liver disease (PLD).

PLD is a group of rare disorders that result from structural changes in the biliary tree development [8]. Genetic mechanisms and/or signaling defects are the main cause of ductal structures that become separated from the biliary tree finally resulting in hepatic cystogenesis [9,10]. Cyst formation in the liver may solely affect the intrahepatic bile ducts (i.e., autosomal dominant polycystic liver disease (ADPLD)) or arise associated with renal cysts [(i.e., autosomal dominant polycystic kidney disease (ADPKD), autosomal recessive polycystic kidney disease (ARPKD), Caroli disease (CD) and CD accompanied by congenital hepatic fibrosis (CHF) in infants, known as Caroli syndrome (CS) [11,12]). Among PLDs, CD patients have an increased risk of developing CCA [13].

The majority of PLD-related genes (i.e., *PRKCSH*, *SEC63*, *ALG8*, *PKD2*, *GANAB*, *SEC61β*) encode for endoplasmic reticulum (ER)-resident proteins involved in the biogenesis, synthesis, maturation, folding and transport of nascent proteins [14]. Therefore, mutations in these genes may compromise ER protein homeostasis, thereby activating the unfolded protein response (UPR) signaling cascades, consisting of sensor (IRE1α, PERK and ATF6) and effector (CHOP, BiP/GRP78 and XBP1) proteins, in order to promote protein folding, ER-associated protein degradation (ERAD) and the activation of prosurvival mechanisms [12].

In the present study, we hypothesized that fibropolycystic-related pathogenesis in *Jnk^∆hepa^* mice might be triggered concomitantly with the activation of the UPR in response to abnormal ER protein homeostasis thus contributing to uncontrolled cholangiocyte proliferation and cell death.

## 2. Materials and Methods

### 2.1. Experimental Models

The generation of hepatocyte-specific *Jnk1/2* knockout mice (*Jnk^Δhepa^*) in a C57BL/6 background was earlier described [5]. Albumin (Alb)-Cre animals were purchased from Charles River (Cerdanyola del Vallés, Barcelona, Spain). By homologous recombination in embryonic stem (ES) cells, mice with a floxed allele of *Jnk1/2* were constructed *Jnk1/2LoxP/LoxP* (*Jnk^f/f^*) according to previously published studies, and Alb-Cre mice were crossed with *Jnk1/2LoxP/LoxP* to generate *Jnk^Δhepa^* mice [5,15,16]. Cre-negative *Jnk^f/f^* mice were used as controls.

Age progression of *Jnk^f/f^ and Jnk^Δhepa^* mice was followed, and animals were fed until sacrifice at Weeks 8, 32, 52 and 72 of age. Endoplasmic reticulum (ER) stress induction using tunicamycin (TM) liquid solution (Merck, Madrid, Spain) was performed by diluting TM in sterile 150 mM Dextrose with a fixed working concentration. Eight-week-old *Jnk^f/f^* and *Jnk^Δhepa^* mice received an IP dosage of 2 mg/Kg TM. Control groups received the same IP injection of dextrose solution. Mice were sacrificed 24 h later. Induction of fibrosis in mice was performed using thioacetamide (TAA; Merck, Madrid, Spain) in drinking water (300 mg/L) for a period of 24 weeks to 8-week-old *Jnk^f/f^* and *Jnk^Δhepa^* mice. Another set of *Jnk^Δhepa^* male mice received 25 mg/kg (i.p.) of diethyl-nitrosamine (DEN) at 14 days of age and from Week 8 until Week 22 were treated with CCl_4_ (0.5 mL/kg, i.p.) twice per week. From Week 18 until Week 22, one group of mice (*n* = 6) was treated with the dual G9a/DNMT1 inhibitor CM272 [17,18] (5 mg/kg, i.p.) daily, while control mice (*n* = 6) received the same volume of PBS. For orthotopic implantation, primary tumors from *Jnk^f/f^* and *Jnk^Δhepa^*—DEN/CCl_4_-treated mice were aseptically isolated and placed at room temperature in DMEM supplemented with 10% FBS plus 50 U/mL penicillin and 50 mg/mL streptomycin and implanted in nude mice, as previously reported [19].

Upon sacrifice, serum was collected from the inferior vena cava, and serum alanine aminotransferase (ALT), aspartate aminotransferase (AST), alkaline phosphatase (AP) and lactate dehydrogenase (LDH) were measured in the Institute of Clinical Chemistry at the University Hospital RWTH Aachen (UKA) using automated analyzers. Liver tissue was collected in liquid nitrogen and kept at −80 °C for protein and RNA extraction. A portion was fixed in 4% PFA for immunohistochemistry (IHC) staining and in TissueTek for immunofluorescence (IF) staining.

Both *Jnk^f/f^* mice and *Jnk^Δhepa^* mice were bred and maintained in the Animal Facility at the School of Biology at UCM, Madrid, under pathogen-free conditions in a temperature and humidity-controlled room with 12 h light/dark cycles and allowed food and water *ad libitum*. Animal work was approved by the Consejería de Medio Ambiente, Administración Local y Ordenación del Territorio (PROEX-125.1/20).

### 2.2. Immunoblot Analysis

According to Bradford’s method, protein concentrations in the whole liver tissue lysate, cytoplasm and mitochondria were measured using Bio-Rad protein analysis reagent following the manufacturer’s instructions [20]. Immunoblotting was performed as described [1] using the following primary antibodies: BIP/GRP78 (CST, Leiden, Netherlands), CHOP (CST), GAPDH (Bio-Rad, Madrid, Spain), sXBP1 (CST), uXBP1 (Abcam, Cambridge, UK), pJNK1 (BIONOVA, Littleton, CO, USA), JNK1 (CST), pJNK2 (BIONOVA), JNK2 (CST), CK19 (Abcam, Cambridge, UK) and αSMA (Merck). GAPDH was used as a loading control. Primary antibodies were detected with anti-mouse (Bio-Rad) or anti-rabbit (Werfen, Barcelona, Spain) IgG antibodies, and signals were developed using Amersham ECL Prime (GE Healthcare, Madrid, Spain). Quantification of immunoblot signals was performed with the Image Lab Software from Bio-Rad Laboratories. Values were normalized to control signals and are provided together with blot images.

### 2.3. Histological Evaluation of Samples

Paraffin-embedded hepatic tissue was sectioned and stained for H&E, Periodic Acid Schiff (PAS) and Sirius Red (SR). Samples were examined by a pathologist blinded who analyzed the degree of liver injury. Immunohistochemistry on paraffin sections was performed. Briefly, liver sections were deparaffinized with xylene and rehydrated with serially descending percentages of ethanol. The sections were then boiled in 10 mM sodium citrate acid buffer (pH = 6) to enhance the availability of the antigen, followed by incubation with 3% H_2_O_2_. Afterward, the sections were transferred to 2.5% horse serum (Palex Medical, Barcelona, Spain) and incubated overnight at 4 °C with CK19 (Abcam), Ki67 (Abcam), SOX9 (Abcam), cleaved caspase-3 (CST), NOTCH1 (Abcam) and MUCIN2 (Santa Cruz, Heidelberg, Germany) antibodies.

The following day, slides were incubated with secondary antibodies (Palex Medical, Madrid, Spain) for 1 h at RT in a humidifying box. The signal was developed with diaminobenzidine (DAB, peroxidase substrate kit) (Palex Medical). The sections were counterstained using hematoxylin and mounted with Roti-Histokit (Quimivita, Barcelona, Spain).

For the immunofluorescence staining, frozen cryosections were incubated with αSMA, Ki67, CD11b (BD Biosciences, Madrid, Spain), CD45 (BD Biosciences) and F4/80 (Bio-Rad) overnight and incubated with fluorescence labeled secondary antibodies (AlexaFluor 488, Fisher Scientific, Madrid, Spain). Slides were then mounted with DAPI (Palex Medical) and imaged using Axio Imager A1 microscope (Carl Zeiss AG, Jena, Germany) and AxioVision software. The cryosections from mouse liver were also incubated with an in situ cell death detection kit (Roche, Madrid, Spain) overnight at 4 °C.

### 2.4. RNA Isolation and Quantitative Real-Time Polymerase Chain Reaction (RT-PCR)

Total cellular RNA was isolated with Trizol (Fisher Scientific). For reverse transcription, 1 µg of total RNA was transcribed using Applied Biosystems™ High-Capacity cDNA Reverse Transcription Kit (Fisher Scientific). Quantitative real-time PCR was carried out by a real-time PCR machine employing Sybr Green PCR Master Mix (Fisher Scientific). The Ct values were extrapolated to a standard curve and data was normalized to the house-keeping gene expression (*Gapdh*). Primers’ sequences are available upon request.

### 2.5. Statistical Analysis

All data were expressed as mean ± standard deviation of the mean. The standard error of the mean (SEM) was calculated from the average of at least 3 independent samples per condition. Statistical significance was determined via using GraphPad Prism 8.0 software (GraphPad Software, CA, USA), followed by a Student’s *t*-test (unpaired, two-tailed test) or via two-way analysis of variance (ANOVA), including Tukey’s multiple comparisons test. Values of *p* < 0.05 were considered significant.

## 3. Results

### 3.1. Hepatocytic Deletion of Jnk1/2 Results in Progressive Fibropolycystic Disease Characterized by Extracellular Matrix Deposition and Inflammation

We first generated knockout animals with conditional deletion of both *Jnk1* and *Jnk2* in hepatocytes (*Jnk^∆hepa^*) (Appendix A). Littermates carrying the respective loxP-flanked alleles but lacking expression of Cre recombinase were used as controls (*Jnk^f/f^*) (Appendix A). Progression of liver disease was evaluated at 8, 32, 52 and 72 weeks of age (Appendix A).

Interestingly, strong presence of fibrillar collagen networks was identified using Sirius Red (SR) staining in tissue sections from aging *Jnk^∆hepa^* mice, mimicking CS in patients [7] (Figure 1A). No signs of PKD were observed (Appendix A). The quantification of type I and III Collagen fibers in the liver parenchyma of these mice showed a clear tendency from 32 weeks of age and significant differences at 52 and 72 weeks of age, compared to *Jnk^f/^^f^* mice (Figure 1B). Alpha-smooth muscle actin (αSMA) protein overexpression was evident at 32 weeks of age in *Jnk^∆hepa^* mice (Figure 1C). Moreover, increased mRNA expression levels of *αSma*, *ColIa1* and *Mmp9* was observed in aging *Jnk^∆hepa^* compared with *Jnk^f/^^f^* livers in most of the time points assessed (Figure 1D).

Synthesis of extracellular matrix (ECM) and activation of myofibroblasts and hepatic stellate cells (HSCs) is often accompanied by the recruitment of leukocytes. Thus, we examined by immunofluorescence (IF) the infiltration of inflammatory cells. The numbers of leukocytes (CD45), macrophages (F4/80) and monocytes (CD11b) were significantly increased in the liver of aging *Jnk^∆hepa^* mice from 32 weeks of age, compared with *Jnk^f/^^f^* littermates (Figure 1E and Appendix A). Additionally, the levels of transcripts of proinflammatory cytokines including TNFα and TGFβ1 were significantly elevated in the livers of 32-week-old *Jnk^∆hepa^* compared with *Jnk^f/^^f^* mice (Appendix A).

### 3.2. Hepatocytic Deletion of Jnk1/2 Promotes Hepatomegaly and Liver Damage

Since hepatobiliary pathology manifested by *Jnk^∆hepa^* mimicked CS to a great extent, we next explored the histological and clinical characteristics of *Jnk^∆hepa^* aging mice. Hepatomegaly was observed from 52 weeks of age, when mice showed a significantly increased liver weight to body weight (LW/BW) ratio compared with *Jnk^f/f^* animals (Figure 2A and Appendix A). Histologically, from the 52-week-old time point, 100% of *Jnk^∆hepa^* mice exhibited cysts across the hepatic parenchyma, accompanied by necrotic areas and the presence of inflammatory cells (Figure 2B and Appendix A).

Next, we measured serum levels of surrogate markers of liver injury in aging *Jnk^∆hepa^* knockout mice. The biochemical analysis revealed that ALT and LDH levels were significantly increased in *Jnk^∆hepa^* mice from 52 weeks of age (Figure 2C,D), indicating that hepatocellular injury might be associated with cyst formation observed in *Jnk^∆hepa^* livers.

Therefore, we subsequently assessed cell death and proliferation using IF and IHC techniques. The TUNEL assay detects DNA breakage that arises during early and late stages of apoptosis [21]. The amount of TUNEL-positive cells per view field was significantly increased in *Jnk^∆hepa^* livers from 52 weeks of age, compared with *Jnk^f/f^* animals, indicating increased apoptotic cell death in the absence of hepatocytic *Jnk1/2* (Figure 2E). Interestingly, detection of the cleaved form of Caspase-3 (CC3) by IHC was observable already from 32 weeks of age (Appendix A). In accordance, a compensatory proliferative response, measured using Ki67-, was evident in *Jnk^∆hepa^* mice livers, reaching statistical significance from 32 weeks of age when compared with *Jnk^f/f^* mice (Figure 2F and Appendix A).

### 3.3. Hepatocytic Deletion of Jnk1/2 Triggers Cystic Hyperproliferation and Cholangiocyte Malignancy

Earlier work demonstrated that hepatocellular JNK deficiency is sufficient for the stimulation of cholangiocyte proliferation and the development of malignancy, resulting in the occurrence of CCA [6]. Therefore, we next studied markers of cystic structures and cholangiocyte proliferation such as CK19 in 52-week-old *Jnk^f/f^* and *Jnk^∆hepa^* mice. Whereas CK19-positive staining was exclusive of small bile ducts in *Jnk^f/f^* mice, *Jnk^∆hepa^* livers showed increased staining in cystic areas as a result of massive cholangiocyte proliferation. This was corroborated by significantly elevated CK19 protein and mRNA expression in these livers (Figure 3A,B). Glutamine synthase (GS) is normally expressed around perivenular areas, as in *Jnk^∆hepa^* livers (Figure 3A, center panel). However, transforming cells express GS. We observed strong positive areas of GS staining are visible in *Jnk^∆hepa^* livers (Figure 3A, center panel). Concomitant with our GS results, Periodic Acid Schiff (PAS) staining, characteristic of mucin-containing tissues, was restricted to cystic areas in *Jnk^∆hepa^* livers (Figure 3A, right panel). Next, we analyzed markers of hepatoblast differentiation. Interestingly, the mRNA expression of *Yap1*, a marker of hepatoblast differentiation [22] and *Jag1* and *Hey1*, genes involved in the Notch signaling pathway [23], were significantly upregulated in 52-week-old *Jnk^∆hepa^* compared with *Jnk^f/f^* mice (Figure 3C–E).

### 3.4. Activation of the Unfolded Protein Response (UPR) Is Associated with Increased Liver Injury in Jnk^∆hepa^ Mice

Hepatic cystogenesis in PLD patients has been recently associated with abnormalities in protein homeostasis in endoplasmic reticulum (ER) [12]. Moreover, severe or prolonged ER stress in epithelial cells is known to result in myofibroblast activation and fibrosis development in different tissues [24]. Therefore, since we detected αSMA overexpression and ECM deposition, indicative of activated myofibroblasts and HSCs, in the livers of *Jnk^∆hepa^* mice from 32 weeks of age, we analyzed the protein levels of the main UPR components in liver tissue of our experimental groups. Interestingly, while no differences in pIREα protein expression were observed between *Jnk^f/f^* and *Jnk^∆hepa^* mice livers (not shown), the UPR effectors BiP/GRP78 and CHOP were strongly overexpressed in 32-week-old *Jnk^∆hepa^* livers (Figure 4A). These findings were associated with increased levels of spliced and total XBP1 protein levels in *Jnk^∆hepa^* compared with *Jnk^f/f^* animals (Figure 4A).

Tunicamycin (TM), a bacterial nucleoside that causes accumulation of unfolded or misfolded proteins in the ER, efficiently induces ER stress and activation of the UPR in the liver in vivo [25,26]. Therefore, we used TM as an experimental tool to directly evaluate the effect of UPR activation as a trigger in the fibrogenic response and hepatocellular injury in *Jnk^∆hepa^* mice. As expected, TM administration resulted in the elevation of UPR effector proteins, including BiP/GRP78, CHOP and XBP1 in *Jnk^f/f^* livers (Figure 4B). However, this response was more intense in *Jnk^∆hepa^* animals (Figure 4B), indicating increased UPR activation in hepatocytic *Jnk*1/2 knockout mice. As indicated by SR staining of liver sections, acute TM dosing triggered a mild fibrogenic response in floxed-control *Jnk^f/f^* mice 24 h after administration. In contrast, this response was much more exacerbated in *Jnk^∆hepa^* livers (Figure 4C and Appendix A). Concomitantly, cell death and compensatory proliferation, measured as TUNEL and Ki-67 staining respectively, were significantly increased in the hepatic parenchyma of *Jnk^∆hepa^* compared with *Jnk^f/f^* animals (Figure 4D,E and Appendix A). Altogether these results show that ER stress and UPR activation are increased in hepatocytic Jnk1/2 knockout mice, and these animals are much more sensitive to ER-mediated parenchymal damage, explaining the increased fibrosis and hepatocellular injury observed in these mice.

### 3.5. Jnk^∆hepa^ Mice Display an Exacerbated Profibrogenic Response

Since *Jnk^∆hepa^* mice are more susceptible to ER stress and UPR activation, resulting in an enhanced acute fibrogenic response, we decided to further explore the response of these mice in the context of chronic liver injury. Chronic supplementation in drinking water with thioacetamide (TAA), a toxin known to induce liver injury and cholangiocarcinogenesis in rodents [27] leads to severe fibrosis/cirrhosis between 16 and 24 weeks in mice [28]. Thus, we treated our experimental groups with TAA for a period of 24 weeks (Appendix A). Interestingly, TAA challenge did not result in further induction of UPR effectors except for CHOP expression in *Jnk^∆hepa^* mice compared with *Jnk^f/f^* littermates (Figure 5A). However, fiber deposition was induced in both experimental groups upon TAA administration, as observed by SR staining (Figure 5B,C). Interestingly, a differential pattern of collagen deposition (SR staining), was observed in the livers of *Jnk^f/f^* and *Jnk^∆hepa^* mice. While TAA-induced bridging fibrosis in *Jnk^f/f^* livers, the hepatic parenchyma of *Jnk^∆hepa^* livers displayed peribiliary-like fibrosis (Figure 5B). αSMA levels were evaluated by IF staining and Western blot (Figure 5D,E). Compared with *Jnk^f/f^* mice, activated fibrotic response area was enhanced in *Jnk^∆hepa^* mice (Figure 5C). Consistently, αSMA protein expression was strongly induced in *Jnk^∆hepa^* livers, particularly after TAA challenge (Figure 5E). These results suggest that loss of *Jnk1/2* function in hepatocytes promotes exacerbated liver fibrosis.

### 3.6. Jnk^∆hepa^ Mice Display Extensive Hepatocellular Injury in Response to TAA

Given that TAA supplementation caused a strong induction of liver fibrosis in mice with hepatocytic deletion of *Jnk1/2*, we next sought to evaluate the hepatotoxic response to TAA in our experimental groups. TUNEL staining was performed to detect cell death. Microphotographs and data quantification indicated that TAA significantly induced cytotoxicity in both *Jnk^f/f^* and *Jnk^∆hepa^* animals. However, apoptotic cell death was exacerbated in *Jnk^∆hepa^* compared with *Jnk^f/f^* livers (Figure 6A,D). Concomitant analysis of cell proliferation using Ki67 indicated that compensatory cell proliferation in response to TAA-induced cell death occurred in both *Jnk^f/f^* and *Jnk^∆hepa^* animals. Nevertheless, this response was significantly elevated in *Jnk^∆hepa^* compared with *Jnk^f/f^* livers (Figure 6B,E). Moreover, the frequency of mitotic figures was significantly increased in TAA-treated *Jnk^∆hepa^* compared with *Jnk^f/f^* livers (Figure 6F).

Histological examination of the livers by H&E staining was performed in both *Jnk^f/f^* and *Jnk^∆hepa^* livers. Hepatocellular damage and infiltration of inflammatory cells were prominent in TAA-treated *Jnk^f/f^* mice, compared with vehicle-treated animals. Lesion areas in TAA-treated *Jnk^f/f^* mice presented a prominent acinar pattern, combined with cytologic atypia (Figure 6C). However, in *Jnk^∆hepa^* mice, cells in the lesion area were characterized by strong mitosis accompanied by multiple ductular dilations. At the same time, the ductular dilations and mitotic figures in the lesion areas of TAA-challenged *Jnk^∆hepa^* mice were significantly different, compared with the structural characteristics of spontaneous cystic dilatations of the ductular tracts in the vehicle-treated group (Figure 6C). Consistently with the enhanced apoptotic and proregenerative (Ki67 labeling) responses found in TAA-treated *Jnk^∆hepa^* mice, increased circulating levels of ALT, AST, AP and LDH, markers of hepatocellular and biliary injury were detected in these animals (Figure 6G–J).

### 3.7. Chronic TAA Administration Triggers Cellular Atypia and Markers of Cholangiocarcinogenesis in Jnk^∆hepa^ Mice

As shown above, loss of *Jnk1/2* in hepatocytes gradually triggers the formation of biliary hamartomas accompanied by the development of ER stress, activation of the UPR, and fibrosis and hepatocellular injury responses, which were enhanced by TAA administration. Since CHF patients have increased risk for CCA [29], we sought to investigate whether carcinogenesis was also enhanced by TAA in our experimental setting. After 24 weeks of TAA administration (Appendix A), no significant differences were observed in the LW/BW ratio between TAA-treated *Jnk^f/f^* and *Jnk^∆hepa^* animals (Appendix A). However, the number and diameter of nodules observed in the liver surface was significantly higher in *Jnk^∆hepa^* mice (Appendix A).

In view of this, we next evaluated by IHC analysis the expression of typical markers of oval cells and biliary epithelium, including CK19, SOX9, and MUCIN2. While CK19, SOX9 and MUCIN2 were prominent in untreated *Jnk^∆hepa^* livers, TAA strongly induced their expression compared with *Jnk^f/f^* animals. Analysis of the mRNA expression of *Yap1* and *Ck19* confirmed the enhanced upregulation of biliary proliferation markers [30] induced in the TAA-treated *Jnk^∆hepa^* mice (Figure 7D,E). Moreover, the percentage of SOX9 and MUCIN2-positive cells per HPF20X were significantly increased in TAA-treated *Jnk^∆hepa^* compared with *Jnk^f/f^* mice (Figure 7F,G). Moreover, immunoblotting for CK19 validated the previous observations (Figure 7H). Altogether, markers of cholangiocyte/BECs were strongly activated after treatment with TAA in *Jnk^∆hepa^* livers.

### 3.8. Hepatotoxin-Challenged Jnk^∆hepa^ Mice Develop Fibrocystic Liver Disease and CCA in Association with a Strong UPR Activation: Therapeutic Potential of an Innovative Epigenetic Inhibitor

So far, our findings reveal that *Jnk^∆hepa^* mice spontaneously develop hepatic histological and molecular traits indicative of liver injury, fibrosis and a remarkable cystogenesis accompanied by early activation of the UPR. This sequence of events can be accelerated and enhanced by the chronic administration of the hepatotoxin and carcinogen TAA, which also hastens the emergence of markers of CCA development, as seen above. We recently described that when *Jnk^∆hepa^* mice are treated with a single dose of the carcinogen diethylnitrosamine (DEN), and then chronically challenged with CCl_4_ (DEN/CCl_4_ model), instead of hepatocellular carcinomas, these animals develop cyst-like structures with molecular features compatible with cholangioma and malignant CCA [1,17], including upregulated expression of NOTCH1, GS and markers of oval cells (*Ck19*, *Sox9*, *Yap1* mRNA), as well as increased proliferation measured as *Pcna* mRNA levels (Appendix A), similar to those observed now in response to TAA. In view of this, and of the recognized risk of CCA development in patients with fibrocystic liver diseases, including CD [13,31,32], it was important to provide conclusive evidence of the malignant nature of the lesions developed in these mice. To this end, liver tissue lesions that emerged in *Jnk^∆hepa^* mice after DEN/CCl_4_ treatment (Appendix A) were resected and orthotopically implanted in nude mice, as previously described [19] (Figure 8A). Mice were followed up and sacrificed 28 weeks after orthoallografts implantation. As shown in Figure 8B, all mice harbored tumoral lesions. Moreover, the engrafted implants displayed histological features resembling those found in TAA and DEN/CCl_4_ treated *Jnk^∆hepa^* mice livers, compatible with malignant CCA. Histopathological examination revealed that lesions were constituted as glandular and canalicular with cuboidal epithelium and moderate cytologic atypia. All tumors show expansive margins of growth (Figure 8C).

We previously showed that by 32 weeks of age, *Jnk^∆hepa^* mice develop spontaneous liver injury and an UPR (Figure 1, Figure 2 and Figure 4). These features were still not evident in 22-week-old mice (Figure 9A). However, when subjected to the DEN/CCl_4_ challenge (as described in Appendix A), 22-week-old *Jnk^∆hepa^* mice developed strong activation of the UPR (Figure 9B) and a potent cystogenic and fibrogenic response (Figure 9C). We recently described that the malignant progression of CCA-like lesions in DEN/CCl_4_ treated *Jnk^∆hepa^* mice can be inhibited by the administration of CM272, a novel epigenetic drug that simultaneously targets the histone methyltransferase G9a and DNA-methyltransferase 1 (DNMT1) [17]. Interestingly, CM272 administration, as described in Appendix A, markedly reduced the expression of UPR effectors (Figure 9B) and significantly attenuated the fibrogenic and cystogenic responses (Figure 9C). Most interestingly, CM272 treatment also reduced the levels of NOTCH1 and MUCIN2 and downregulated *Yap1, Jag1* and *Hey1* mRNA transcripts, which were induced in the livers of DEN/CCl_4_-treated *Jnk^∆hepa^* mice (Appendix A).

## 4. Discussion

Fibropolycystic disease is an umbrella term that comprises a spectrum of conditions of the intrahepatic bile ducts, characterized by different histological and clinical findings within the liver and other organs.

We were the first to report that mice with disruption of JNK1/2, hepatocyte specific for JNK1, and systemic JNK2 deletion exhibited bile duct hyperplasia at late stages of age progression. Moreover, combined genetic JNK1- and siRNA-JNK2-mediated hepatocyte-specific deletion also caused hepatic cystogenesis [1]. In the present study, we generated *Jnk^∆hepa^* mice by combining Alb-Cre with *Jnk1Jnk2*LoxP mice. The cyst phenotype has been also observed with the Alfp-Cre promoter, revealing that progressive biliary cysts development is independent of the Cre line [7].

Interestingly, our results indicated that the observed phenotype was strongly associated with elevated collagen synthesis and extracellular matrix deposition with activation of HSCs. This is a relevant observation, since patients with Caroli syndrome (CS) present hepatic fibrosis, and therefore, the *Jnk^∆hepa^* model might be relevant for this relatively understudied disease.

CS is usually diagnosed in early infancy or during childhood, with an estimated incidence rate of 1 in 10,000 to 20,000 live births [33]. Apart from biliary changes, patients with CS exhibit portal fibrosis and inflammation. *Jnk^∆hepa^* mice from 32 weeks of age and after showed prominent periportal and/or bridging fibrosis, as well as infiltration of immune cells. The phenotype of JNK-hepatocyte knockout mice was characterized by spontaneous liver injury in all 52- and 72-week-old mice. At one year of age, *Jnk^∆hepa^* animals had significantly increased serum transaminases, LW/BW ratio and exhibited prominent cystogenesis. Moreover, cell death and compensatory proliferation were remarkable in *Jnk^∆hepa^* mice. However, the nature of cell death remains unclear. In order to understand whether apoptosis or necrosis contributed to biliary proliferation of *Jnk^∆hepa^* mice, Muller and colleagues [7] additionally deleted Caspase-8, Mlkl and Ripk1 by creating triple knockout mice. Only deletion of Ripk1 prevented cyst formation. Unfortunately, the interaction and functional relevance of Ripk1 and Jnk in this process need to be further studied. However, in the same study, the authors detected RIPK1 expression in biliary epithelial cells of cystically dilated bile ducts of patients with Caroli disease/syndrome (with additional congenital hepatic fibrosis) [7], suggesting that the regulation of JNK1/RIPK1 signaling could be a potential therapeutic avenue for PLD patients.

Emerging evidence implicates ER stress and UPR signaling in a variety of profibrotic mechanisms in individual cell types. For instance, in epithelial cells, ER stress can induce a profibrotic microenvironment by promoting cell death and activating inflammatory signaling pathways and inducing production of profibrotic mediators that promote fibroblast proliferation and myofibroblast differentiation [24].

The ER is a major intracellular organelle that performs multiple physiological functions including protein folding, post-translational modifications, biosynthesis of fatty acids and sterols, detoxification of xenobiotics, and the storage of intracellular Ca^2+^ [34]. Upon exposure to potential stressors such as drugs, the ER initiates the UPR to restore homeostasis [35]. Specifically important is the ER protein homeostasis, also called proteostasis. A recent publication reported that mutations in genes related to PLD compromise ER proteostasis [12]. While we did not observe changes in sensor proteins of the UPR, including IRE1 or PERK, we observed overexpression of BiP/GRP78 and XBP1 proteins, indicating activation of the UPR effectors in the absence of hepatocytic JNK1/2.

A standard experimental dose of TM (2 mg/kg) induced substantial ER stress in wildtype mice, while robust and exacerbated UPR activation was observed in *Jnk^∆hepa^* mice after TM challenge, as revealed by overexpression of BiP, CHOP and spliced XBP1, and triggered a fibrogenic response related to cellular injury and compensatory proliferation. To further relate the profibrogenic response to UPR activation and spontaneous liver injury in the absence of hepatocytic JNK1/2, we next challenged *Jnk^∆hepa^* mice with the profibrotic drug TAA, which causes peribiliary fibrosis [36]. Interestingly TAA administration triggered a more robust UPR activation in *Jnk^∆hepa^* animals associated with increased activation of αSMA expression. Moreover, TAA triggered cellular atypia and mitotic figures, as well as centrilobular necrosis in floxed mice. *Jnk^∆hepa^* animals exhibited strong cholangiocellular injury, as well as significantly altered makers of liver injury.

CS and other ductular plate malformations are known to be risk factors for hepatobiliary neoplasms and more frequently for CCA [37]. Indeed, a recent multicenter study confirmed a CCA rate of 7.3% in a large series of CD and CS patients [32]. Previous work by the group of Sabio [6] reported that changes in bile acid metabolism of *Jnk^∆hepa^* mice may contribute to cholangiocyte proliferation and hepatoblast maturation, causing bile duct hyperplasia and cholangiocyte injury, which leads to CCA development at late stages. Others did not detect CCA-neoplastic or -dysplastic areas at least at 52 weeks of age [7]. In our hands, progressive expression of markers of hepatoblast differentiation and Notch signaling, which, together with Yap, is a key pathway in cholangiocarcinogenesis [38,39], were detected already in 52-week-old *Jnk^∆^^hepa^* mice in the absence of neoplastic lesions on their liver surface. Importantly, nodules were visible in the surface of TAA- and DEN/CCl_4_ *Jnk^∆hepa^*-treated mice livers, as well as the presence of markers of biliary epithelium atypia and cholangiocarcinogenesis, such as the activation of Notch1 and Yap pathways. Malignancy of these tissues was confirmed by orthotopic implantation in athymic NGS mice, also supporting the metastatic potential of the lesions found in chronically injured *Jnk^∆hepa^* mice livers. These results are in agreement with our previous publication using NEMO*^∆hepa^* mice [1], altogether indicating that *Jnk^∆hepa^* livers are sensitive toward CCA development. Interestingly, we also observed that specific inhibition of the epigenetic enzymes G9a and DNMT1, upregulated in human CCA [17] in DEN/CCl_4_-challenged *Jnk^∆^^hepa^* mice, markedly reduced the activation of the UPR, indicating the attenuation of the ER stress response. Mechanistically, it is unlikely that G9a inhibition directly results in the abolition of ER stress, as the level of H3K9me3, the chromatin repressive mark to which G9a contributes, is reduced on the promoters of *Chop* and *BiP/Grp78* genes concomitant with their upregulation during ethanol-induced chronic liver injury and ER stress in mice [40]. Inflammatory mediators play a central role in triggering ER stress and the UPR in liver injury and chronic liver diseases [41]. The strong anti-nflammatory effect mediated by G9a/DNMT1 inhibition that we previously described in DEN/CCl_4_
*Jnk^∆^^hepa^* mice and other mouse models of chronic liver injury [17,42] might contribute to explain the attenuation of the UPR by CM272 treatment found in the present study. Noteworthy, here, we also observed that CM272 treatment concomitantly reduced the activation of the Notch pathway. This response is consistent with the stimulatory effect of G9a on Notch signaling previously described in endothelial precursor cells [43]. Notch inhibition, together with a potent TGFβ antagonism as we reported [42], may underlie the therapeutic effects of CM272 on liver fibrocystic disease and carcinogenic progression [31,44]. Nevertheless, the detailed mechanisms of CM272-mediated inhibition of the Notch pathway need to be further explored.

## 5. Conclusions

Our study links ER stress and fibrosis with cholangiocellular injury and cell death in *Jnk^∆hepa^* mice, which are sensitive to CCA development. Therefore, the *Jnk^∆hepa^* model can be a relevant experimental tool for the study of fibropolycystic liver diseases including CS. Our work also identifies potential therapeutic strategies for a group of diseases lacking effective medical treatments.

## Figures and Tables

**Figure 1 cancers-14-00078-f001:**
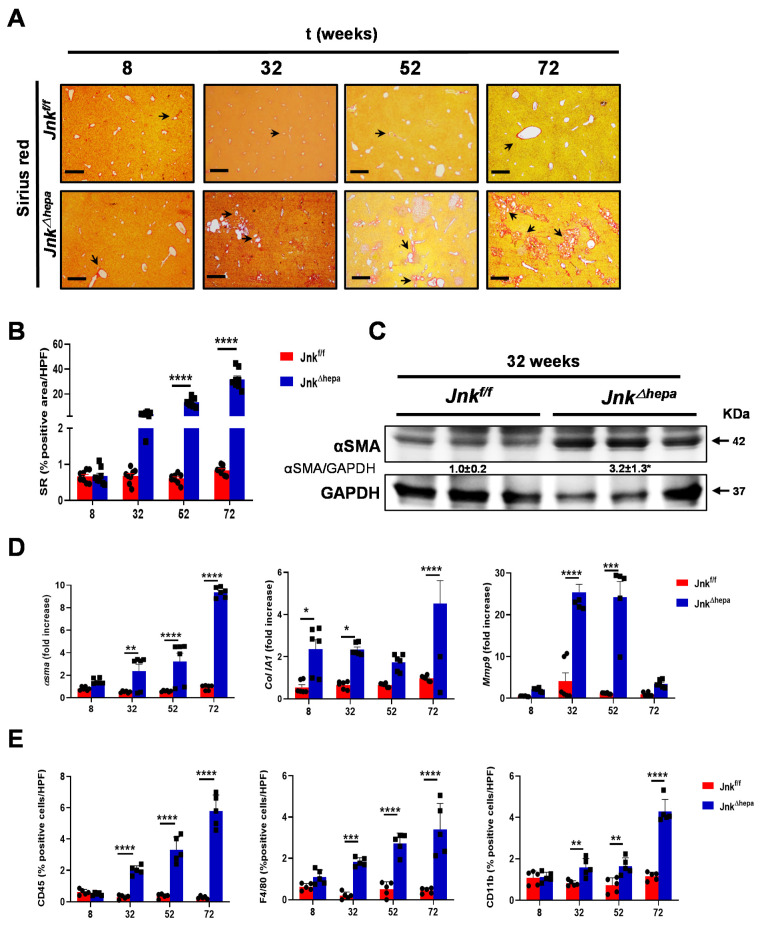
Fibropolycystic disease in aging *Jnk^∆hepa^* mice is characterized by extracellular matrix deposition and inflammation. (**A**) Fibrosis was evaluated by SR staining in 8- to 72-week-old *Jnk^f/f^* and *Jnk^Δhepa^* mice. Scale bars, 500 μm. (**B**) Quantification of SR areas was performed using ImageJ. Protein and mRNA expression was analyzed for *αSm*a (**left panel**), *ColIA1* (**center panel**) and *Mmp9* (**right panel**) using Western Blot (**C**) and qRT-PCR (**D**), respectively (* *p* < 0.05; intergroup significance). (**E**) Quantification of positive cells from IF microphotographs of CD45 (**left panel**), F4/80 (**center panel**) and CD11b (**right panel**) is shown in 8- to 72-week-old *Jnk^f/f^* and *Jnk^Δhepa^* mice. Data are shown as the mean ± SEM and graphed, separately (*n* = 6 mice per group) (* *p* < 0.05; ** *p* < 0.01; *** *p* < 0.001; **** *p* < 0.0001).

**Figure 2 cancers-14-00078-f002:**
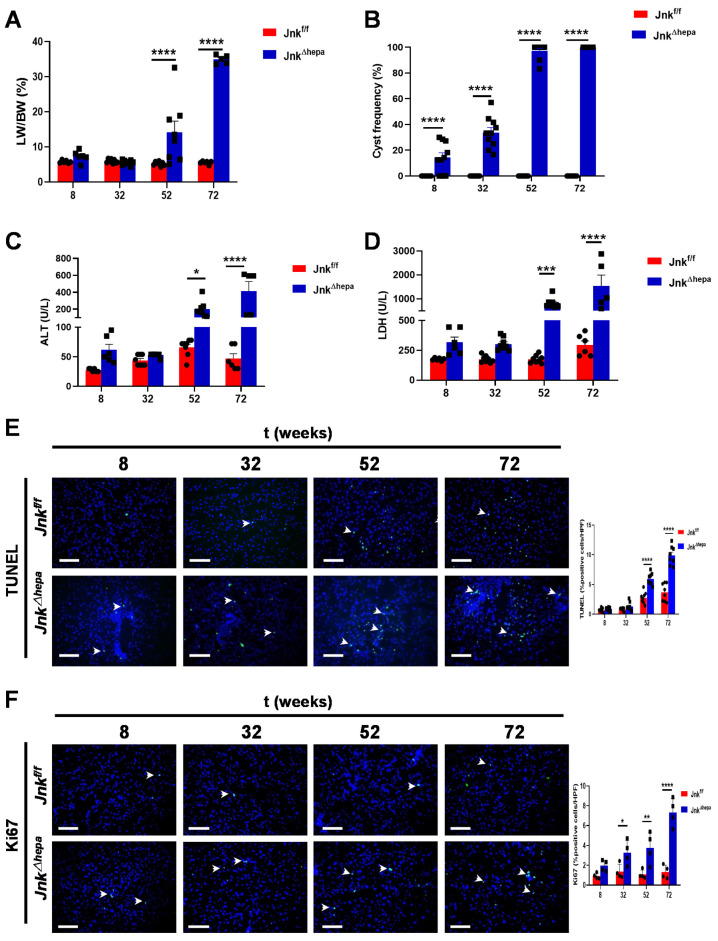
Hepatic deletion of *Jnk1/2* promotes hepatomegaly and liver damage. Eight- to 72-week-old male *Jnk^f/f^* and *Jnk^Δhepa^* mice were analyzed. (**A**) Representation of the LW/BW ratio. (**B**) Liver cysts were counted in the H&E liver tissue staining and represented. Serum ALT (**C**) and LDH (**D**) levels were analyzed. (**E**) Representative microphotographs of TUNEL stainings from *Jnk^f/f^* and *Jnk^Δhepa^* mice liver tissues collected from 8 to 72 weeks of age and quantification of TUNEL-positive cells/HPF. Scale bars, 50 µm. (**F**) Representative microphotographs of Ki67 IF and quantification of Ki67-positive cells/HPF from the same mice. Scale bars, 50 µm. Data are shown as the mean ± SEM and graphed, separately (*n* = 6–8 mice per group) (* *p* < 0.05; ** *p* < 0.01; *** *p* < 0.001; **** *p* < 0.0001).

**Figure 3 cancers-14-00078-f003:**
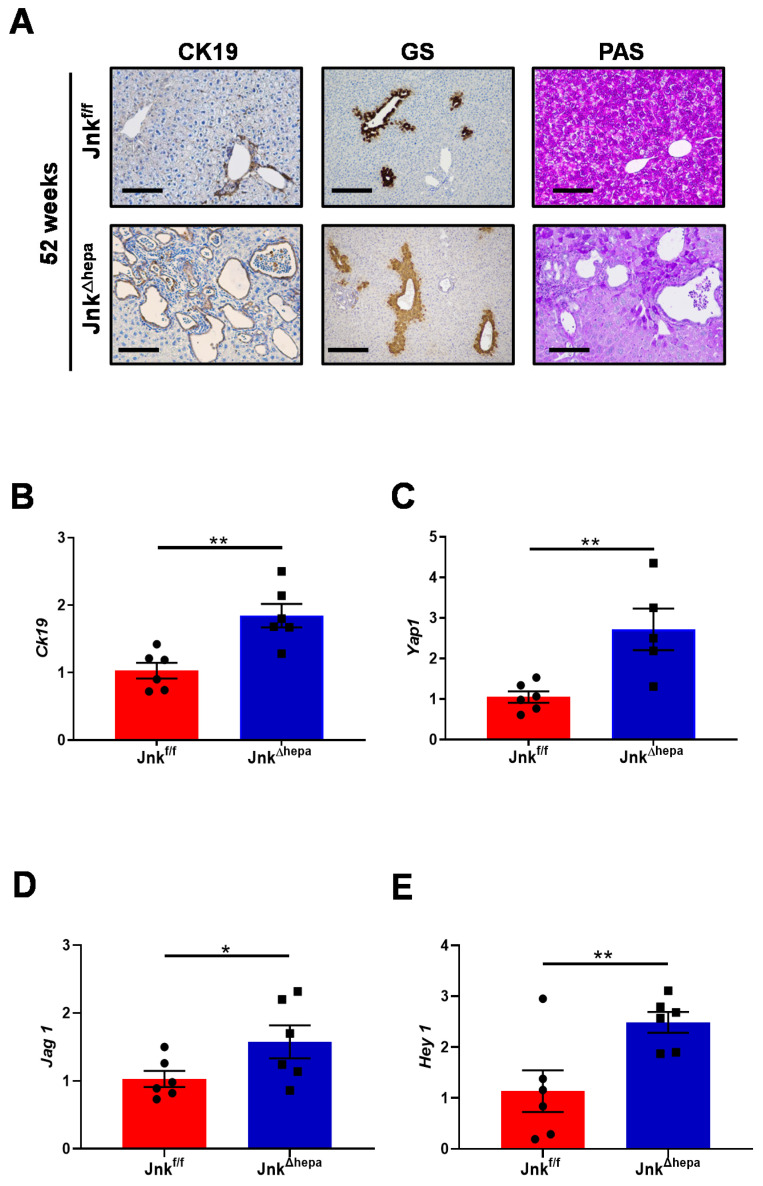
End-stage disease in liver injury of 52-week-old *Jnk^∆hepa^* mice. (**A**) Expression of CK19 (left panel), Glutamine synthase (GS) and Periodic Acid Schiff (PAS) were performed in paraffin sections from 52-week-old *Jnk^f/f^* and *Jnk^Δhepa^* mice livers. Scale bars, 100 μm. (**B**–**E**) Expression of *Ck19*, *Yap1*, *Jag1* and *Hey1* was analyzed and graphed. The data were normalized for the amount of *Gapdh* mRNA in each sample. Data were represented as the mean ± SEM and graphed, separately (*n* = 6 mice per group, * *p* < 0.05; ** *p* < 0.01).

**Figure 4 cancers-14-00078-f004:**
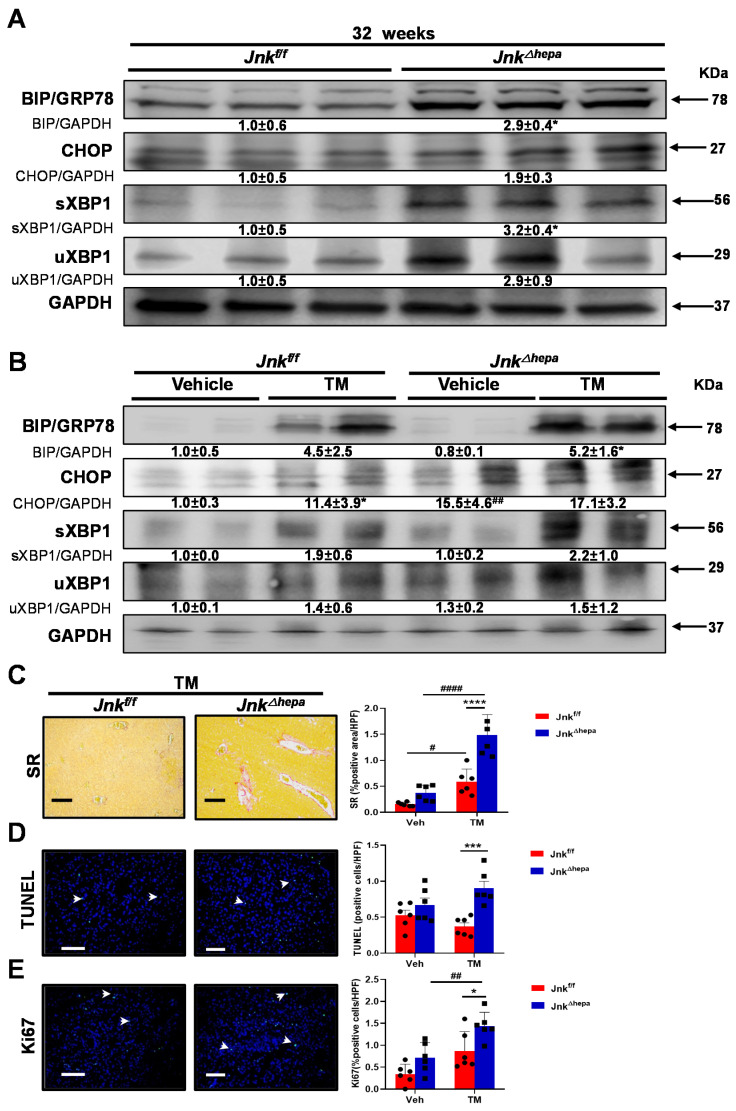
Activation of the unfolded protein response (UPR) triggers increased liver injury in *Jnk^∆hepa^* mice. (**A**) The expression of BiP/GRP78, CHOP, spliced XBP1 (sXBP1) and unspliced XBP1 (uXBP1) was evaluated in 32-week-old *Jnk^f/f^* and *Jnk^∆hepa^* mice using Western blot. (**B**) The expression of the same proteins was measured in Tunicamycin (TM)-challenged *Jnk^f/f^* and *Jnk^∆hepa^* mice. Numbers denote molecular weight (KDa) of proteins. GAPDH served as loading control. #/* *p* < 0.05 intra- and intergroup significance, respectively. (**C**) Representative Sirius Red (SR) stainings of liver tissue from the indicated mice after TM treatment. Scale bars, 500 μm. (**D**) TUNEL staining was performed to assess apoptotic cell death in the same samples by IF. Scale bars, 50 μm. (**E**) Ki67 was used to measure cell proliferation after TM treatment in the experimental groups. Scale bars, 50 μm. Data were represented as the mean ± SEM and graphed (*n* = 3–6 mice per group;*/# *p* < 0.05; ## *p* < 0.01; *** *p* < 0.001; ****/#### *p* < 0.0001).

**Figure 5 cancers-14-00078-f005:**
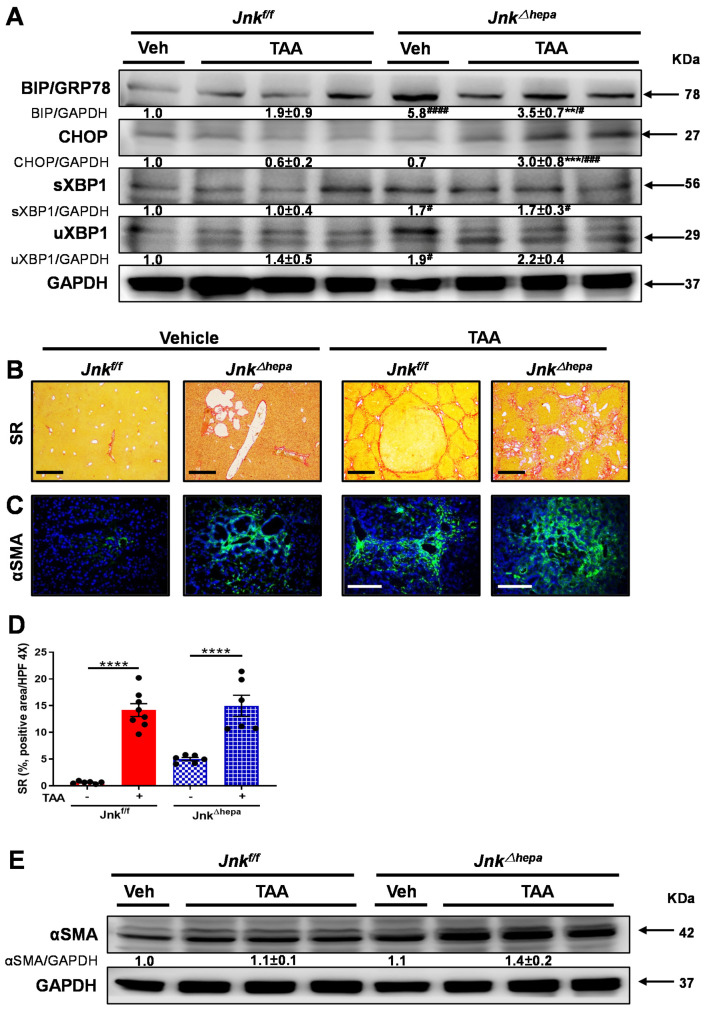
TAA challenge exacerbates liver fibrogenesis in mice with hepatocytic deletion of *Jnk1/2*. (**A**) The expression of BiP/GRP78, CHOP, spliced XBP1 (sXBP1) and unspliced XBP1 (uXBP1) was evaluated in the livers of *Jnk^f/f^* and *Jnk^∆hepa^* mice treated or not with TAA using Western blot. Numbers denote molecular weight (KDa) of proteins. GAPDH served as loading control. (**B**) Representative Sirius Red (SR) stainings of liver paraffin sections from control and TAA-treated *Jnk^f/f^* and *Jnk^∆hepa^* mice (*n* = 6-8 mice per group). Scale bars, 500 μm. (**C**) Expression of αSMA protein was assessed via IF staining. Scale bars, 50 μm. (**D**) Positive area of fibrosis was calculated by ImageJ with microphotographs of SR staining. Data were represented as the mean ± SEM and graphed. (**E**) Expression of αSMA was analyzed by Western blot in the indicated groups of mice. Numbers denote molecular weight (KDa) of proteins. GAPDH served as loading control. (# *p* < 0.05; ** *p* < 0.01; ***/### *p* < 0.001; ****/#### *p* < 0.0001).

**Figure 6 cancers-14-00078-f006:**
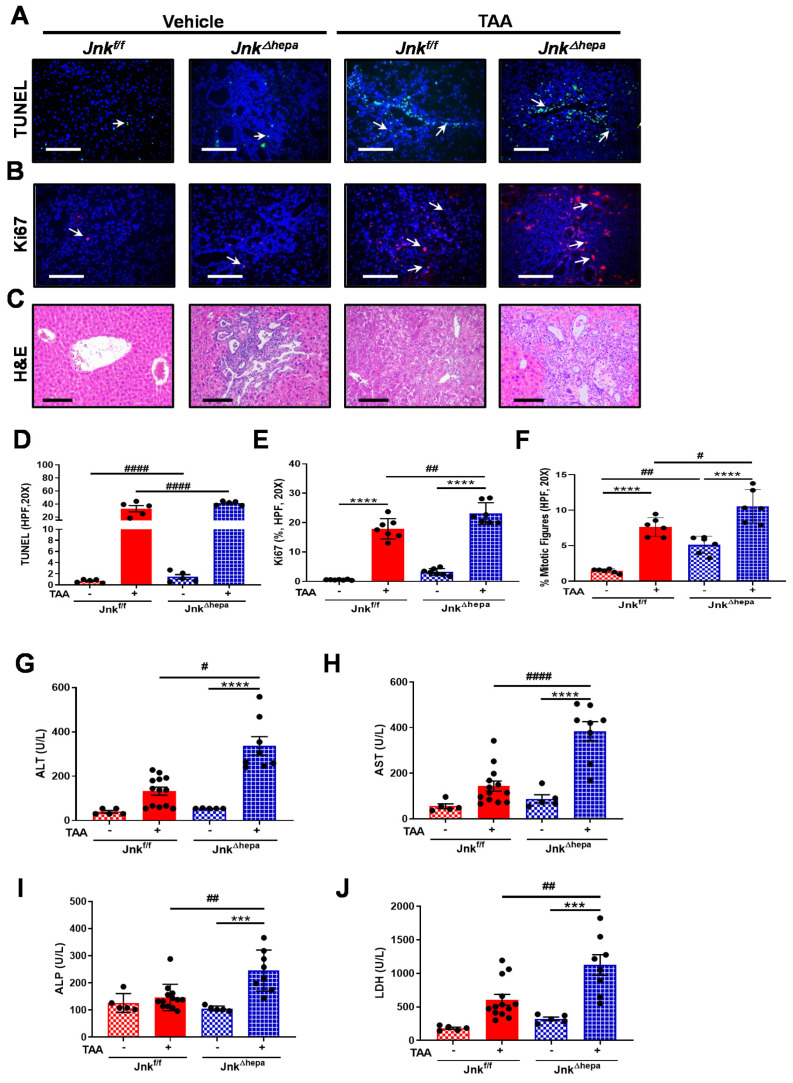
*Jnk^∆hepa^* mice display extensive hepatocellular and cholangiocellular injury in response to TAA. (**A**) Cell death was evaluated using TUNEL staining of liver tissue sections from TAA-treated *Jnk^f/f^* and J*nk^Δhepa^* mice. Arrows (→) indicate TUNEL positive cells. Scale bars, 50 μm. (**B**) Expression of Ki67 was evaluated by IF staining. Scale bars, 50 μm. (**C**) Liver histopathology was analyzed by H&E staining in each experimental group. Positive TUNEL (**D**)**,** Ki67 cells (**E**) and frequency of mitotic figures (**F**) were analyzed by ImageJ and graphed. (**G**–**J**) Analysis of serum levels of ALT, AST, ALP and LDH was assessed. Data were represented as the mean ± SEM and graphed (*n* = 5–13 mice per group; # *p* < 0.05; ## *p* < 0.01; *** *p* < 0.001; ****/#### *p* < 0.0001).

**Figure 7 cancers-14-00078-f007:**
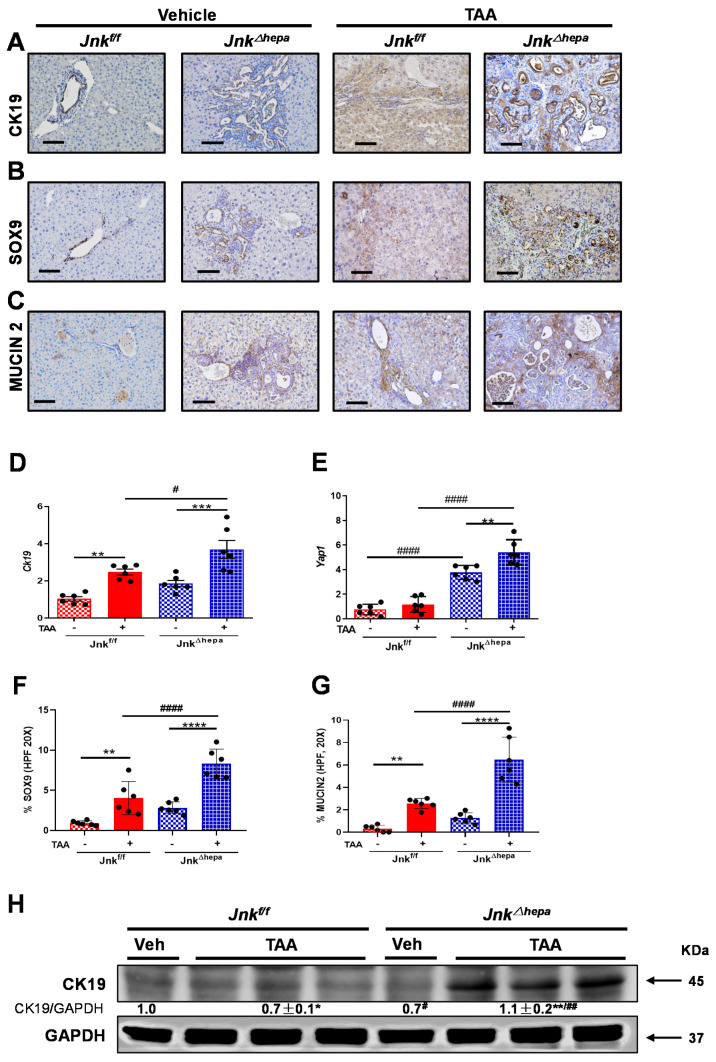
**Figure 7**. Markers of cholangiocarcinogenesis in *Jnk^Δhepa^* mice livers upon chronic TAA challenge. (**A**–**C**) Representative IHC staining for CK19, SOX9 and MUCIN2. Scale bars, 100 μm. (**D**,**E**) mRNA expression of *Yap1* and *Ck19*, was evaluated by qRT-PCR. Quantification of Sox9 and Mucin2 positive cells/HPF (20X) was performed in the same livers. Percentage of SOX9 (**F**) and MUCIN2 (**G**) positive cells was performed in HPF (20X) in liver paraffin sections and graphed. (**H**) Expression of CK19 protein was assessed via Western blot. Numbers denote the molecular weight (KDa) of proteins. GAPDH served as loading control. (*n* = 6; */# *p* < 0.05; **/## *p* < 0.01; *** *p* < 0.001; ****/#### *p* < 0.0001).

**Figure 8 cancers-14-00078-f008:**
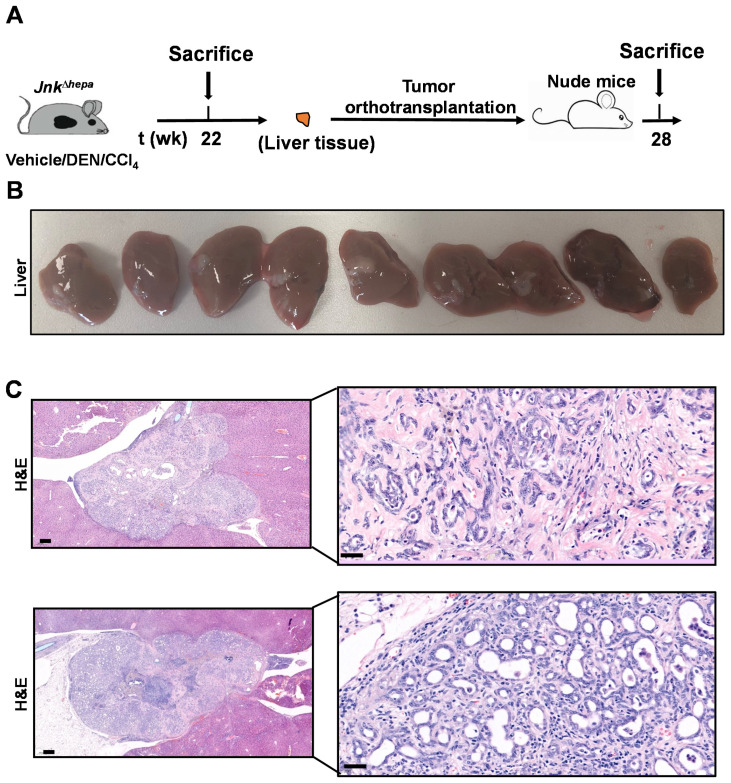
Lesions developed in DEN/CCl_4_ treated *Jnk*^∆hepa^ mice are malignant and display CCA histological features. (**A**) Lesions developed in DEN/CCl_4_ treated *Jnk*^∆hepa^ mice were excised and orthotopically implanted in nude mice (*n* = 9), which were sacrificed and analyzed 28 weeks later. (**B**) Macroscopic aspect of livers from recipient mice after orthotopic implantation of lesions from DEN/CCl_4_ treated *Jnk*^∆hepa^ animals. (**C**) H&E stainings of representative tissue sections of orthoallografts growing in the livers of recipient animals. Scale bars, 50 (**left panels**) and 200 (**right panels**) µm, respectively.

**Figure 9 cancers-14-00078-f009:**
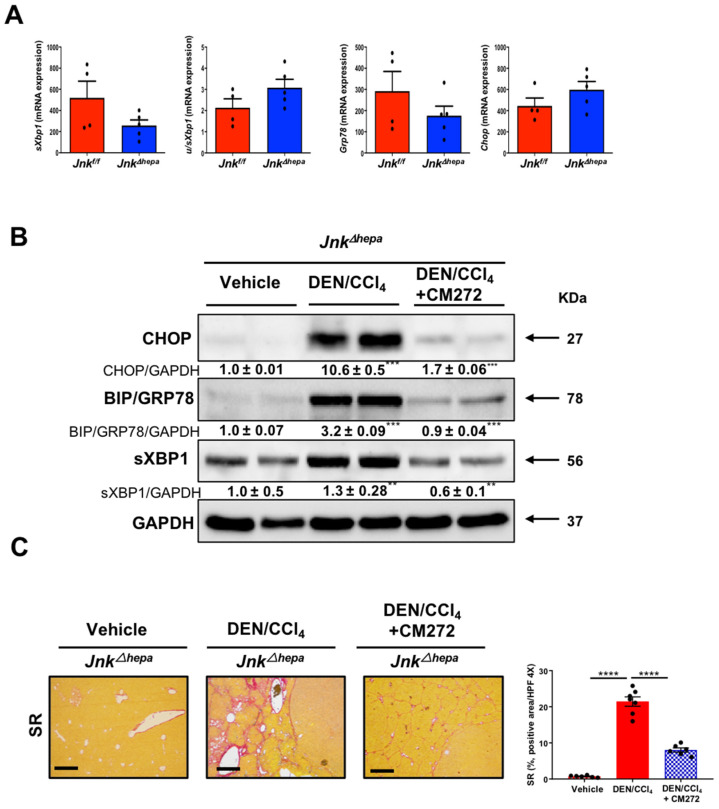
DEN/CCl_4_ challenge in *Jnk^Δhepa^* mice results in the activation of the unfolded protein response (UPR), cystogenesis and fibrosis in the liver: inhibitory effect of targeting the epigenetic G9a/DNMT1 complex. (**A**) mRNA expression levels of *sXbp1*, *uXbp1*, *Bip/Grp78* and *Chop* evaluated by qRT-PCR in the livers of 22-week-old *Jnk^fl/fl^* and *Jnk^Δhepa^* mice. (**B**) Expression of CHOP, BIP/GRP78 and sXBP1 proteins as assessed by Western blot in *Jnk^Δhepa^* control mice (Vehicle), DEN/CCl_4_-challenged mice and DEN/CCl_4_-challenged mice treated with the G9a/DNMT1 inhibitor CM272. Numbers denote the molecular weight (KDa) of proteins. GAPDH served as loading control. Representative images are shown. Bands were quantitated and Vehicle vs. DEN/CCl_4_ samples, and DEN/CCl4 vs. DEN/CCl_4_ + CM272 groups were respectively compared. (**C**) Fibrosis was evaluated by Sirius Red (SR) staining in *Jnk^Δhepa^* mice treated as indicated. Scale bars, 500 μm. Quantification of SR areas using ImageJ was performed. Data are represented as the mean ± SEM (*n* = 5–7 mice per group, ** *p* < 0.01; *** *p* < 0.00,**** *p* < 0.0001).

## Data Availability

The data presented in this study are available in this article (and Appendix A).

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
