# Peer review of "Activation of the Unfolded Protein Response (UPR) Is Associated with Cholangiocellular Injury, Fibrosis and Carcinogenesis in an Experimental Model of Fibropolycystic Liver Disease"

_cancers, 2021, doi:10.3390/cancers14010078_

Round 1

Reviewer 1 Report

This study explores the molecular mechanisms of fibropolycystic disease using the new mouse model, hepatocyte-specific deletion of both Jnk1 and Jnk2 genes, generated by the authors. The work is performed mainly on a high technical level, and the conclusions made by the authors are supported by their data. However, some questions to the authors should be answered, and multiple technical mistakes in the text of the article should be corrected.

Major comments

  1. The images of the GAPDH protein in the immunoblot shown in Fig. 5A are overexposed. Thus, a quantitation of the tested proteins normalized to GAPDH in this figure is not reliable. The same problem exists in the Fig. 7F.

  1. In the description of the results of TAA treatment (Fig. 6) authors mentioned that “…in JnkΔhepa mice, cells in the lesion area were characterized by strong mitosis…”. Why authors did not count directly the mitotic figures on the stained slides of liver tissue? – It could be, that the difference in the frequency of mitotic figures between JNK-KO and WT livers would be more significant than the difference in Ki67 marker frequency.
  2. In the Results section, while describing the results of IHC staining for SOX9 & MUCIN2 presented in the Fig. 7, authors declare that “…TAA strongly induced their expression compared with Jnkf/f animals”. Quantitation of the IHC slides is required to confirm this statement.

  1. The difference in the total number of nodules following chronic TAA treatment between JNK-KO and control mice is statistically significant, but not very prominent (Suppl. Fig. 7F). It is interesting to compare the diameters of these nodules: whether in JNK-KO livers the average diameter of nodules is significantly larger than in control livers (as it can be seen, for example, on the pictures of TAA treatment in Suppl. Fig. 7B)?

  1. Concerning the orthoallografts implantation (results presented in Fig. 8): whether this experiment was performed with the control JNK fl/fl livers as well? Could it be that the control livers will produce in this experiment the same (or similar) results?

  1. Suppl. Fig. 8A is identical to Suppl. Fig. 9A and does not describe the experiment presented in Fig. 8.

Minor comments

  1. In the Fig. 1A, the bottom image (JNK1,2-KO liver) at the age 52 weeks is not a representative one: it is very similar to the upper image (for WT liver), while, according to the quantitation (Fig. 1B), it should have more than 10-fold higher level of SR staining than WT liver (it should be more similar to the bottom image at the age 72 weeks).
  2. In the legend to Fig. 1, Figs. 1C & 1D are described as Fig. 1C, and the Fig. 1E is described as Fig. 1D).
  3. In the Results section, when describing Fig. 3, the age of mice should be shown in the text (it is shown in Fig. 3A only, and not in the legend to Fig. 3).
  4. Lane 324: “…(TAA), a toxic known to induce liver injury…” – probably, should be “a toxin”, or “a toxic substance/compound…”.
  5. In the legend to Fig. 5, legends to Fig. 5C & 5D are switched.
  6. Lane 353 : “…apoptotic cell death was exacerbated JnkΔhepa compared with Jnkf/f livers (Figure 6A, D” – “in” should be inserted after “exacerbated”.
  7. Lane 369: “…hepatocellular and biliary were detected” – insert “injury”.
  8. Lane 385: “LW/BW ratio between TAA-treated JnkΔhepa and JnkΔhepa animals” – one group should be Jnk fl/fl.
  9. Lanes 437 & 443: should it be indeed “Suppl. Fig. 8A”, or “Suppl. Fig. 9A”?
  10. Lane 447: instead of “(Suppl. Fig. 9A and B)” it seems, there should be “(Suppl. Fig. 9B and C)”.

Author Response

Reviewer#1

This study explores the molecular mechanisms of fibropolycystic disease using the new mouse model, hepatocyte-specific deletion of both Jnk1 and Jnk2 genes, generated by the authors. The work is performed mainly on a high technical level, and the conclusions made by the authors are supported by their data. However, some questions to the authors should be answered, and multiple technical mistakes in the text of the article should be corrected.

Major comments

  1. The images of the GAPDH protein in the immunoblot shown in Fig. 5A are overexposed. Thus, a quantitation of the tested proteins normalized to GAPDH in this figure is not reliable. The same problem exists in the Fig. 7F. 

Reply: We apologize for the overexposure. The images were taken in February 2020 and stored in an old computer. In the revised version of the manuscript we have tried to improve the exposure of the GAPDH of both Figs. 5A, and 7F. Additionally, we provide in Figure 1 only for Reviewers, the original membranes.

Figure 1 Only for Reviewers. Original membranes were developed and photographed, and the images were saved as tiff.

  1. In the description of the results of TAA treatment (Fig. 6) authors mentioned that “…in JnkΔhepa mice, cells in the lesion area were characterized by strong mitosis…”. Why authors did not count directly the mitotic figures on the stained slides of liver tissue? – It could be, that the difference in the frequency of mitotic figures between JNK-KO and WT livers would be more significant than the difference in Ki67 marker frequency.

Reply: This is a great point. We have asked our expert pathologist to blindly quantify mitotic figures in all mouse groups. In the revised version of the manuscript, we have included the frequency of mitotic figures which is significantly increased in TAA-treated Jnk∆hepa compared with Jnkf/f livers (Figure 6F).

  1. In the Results section, while describing the results of IHC staining for SOX9 & MUCIN2 presented in the Fig. 7, authors declare that “…TAA strongly induced their expression compared with Jnkf/f animals”. Quantitation of the IHC slides is required to confirm this statement.

Reply: We are grateful for this suggestion. We have quantified the % of SOX9 and MUCIN2 positive cells/HPF. Interestingly, the percentage of SOX9 and MUCIN2-positive cells per HPF20X was significantly increased in TAA-treated Jnk∆hepa compared with Jnkf/f mice. These data have been included in the revised version of the manuscript as Fig. 7F-G.

  1. The difference in the total number of nodules following chronic TAA treatment between JNK-KO and control mice is statistically significant, but not very prominent (Suppl. Fig. 7F). It is interesting to compare the diameters of these nodules: whether in JNK-KO livers the average diameter of nodules is significantly larger than in control livers (as it can be seen, for example, on the pictures of TAA treatment in Suppl. Fig. 7B)?

Reply: Thank you for this suggestion. In the revised version of the manuscript, we have included the diameters of the nodules (Suppl Fig. 7G). Interestingly, the diameter of nodules on the liver surface was significantly higher in Jnk∆hepa compared with Jnkf/f animals, after TAA treatment.

  1. Concerning the orthoallografts implantation (results presented in Fig. 8): whether this experiment was performed with the control JNK fl/fl livers as well? Could it be that the control livers will produce in this experiment the same (or similar) results?

Reply: This is an isightful comment. As we have earlier shown [1], treatment of wildtype Jnkf/f mice with DEN/CCl4 triggers features of hepatocellular carcinoma (HCC) development. However, Jnk∆hepa mice treated with DEN/CCl4 develop cholangiocarcinoma (CCA) instead [2]. Thus, the main goal of this transplantation experiment was to fully evaluate the malignancy of these lesions and whether Jnk∆hepa – liver tissues were capable of recapitulating CCA histological characteristics in nude mice. Our results indicate that the engrafted implants displayed histological features resembling malignant CCA, as evaluated by an expert pathologist, Dr. August Vidal.

  1. Suppl. Fig. 8A is identical to Suppl. Fig. 9A and does not describe the experiment presented in Fig. 8.

 Reply: As the Reviewer mentions both Suppl. Fig. 8A and 9A correspond to the same mouse model. However, Suppl. Fig. 8A is a representation of the DEN/CCl4 model alone, whilst Suppl. Fig. 9A refers to the same model plus the treatment with CM272. This has now been corrected both in the text, and in the Suppl. Material section.

Minor comments

  1. In the Fig. 1A, the bottom image (JNK1,2-KO liver) at the age 52 weeks is not a representative one: it is very similar to the upper image (for WT liver), while, according to the quantitation (Fig. 1B), it should have more than 10-fold higher level of SR staining than WT liver (it should be more similar to the bottom image at the age 72 weeks).

Reply: Thank you for raising this point. We have replaced the image for a more representative picture.

  1. In the legend to Fig. 1, Figs. 1C & 1D are described as Fig. 1C, and the Fig. 1E is described as Fig. 1D).

Reply: We apologize for this mistake. We have corrected the text in the Figure legend. While Fig. 1C is the immunoblot of SMA, Fig. 1D shows the mRNA expression analyzed for αSma (left panel), ColIA1 (center panel) and Mmp9 (right panel). Additionally, Fig. 1E refers quantification of positive cells from IF microphotographs of CD45 (left panel), F4/80 (center panel) and CD11b (right panel) in 8 to 72 weeks-old Jnkf/f and JnkΔhepa mice.

  1. In the Results section, when describing Fig. 3, the age of mice should be shown in the text (it is shown in Fig. 3A only, and not in the legend to Fig. 3).

Reply: Thank you for this observation. We have included the age of the mice ¨..52 weeks-old Jnkf/f and Jnk∆hepa mice..¨ both in the text and in the figure legend of the revised version of the manuscript.

  1. Lane 324: “…(TAA), a toxic known to induce liver injury…” – probably, should be “a toxin”, or “a toxic substance/compound…”.

Reply: We have replaced ¨toxic¨ with ¨toxin¨.

  1. In the legend to Fig. 5, legends to Fig. 5C & 5D are switched.

Reply: We apologize for this mistake, which is now amended in the revised version of the manuscript.

  1. Lane 353 : “…apoptotic cell death was exacerbated JnkΔhepa compared with Jnkf/f livers (Figure 6A, D” – “in” should be inserted after “exacerbated”.

Reply: Thank you, we have added ¨in¨.

  1. Lane 369: “…hepatocellular and biliary were detected” – insert “injury”.

Reply: Thank you, we have added ¨injury¨.

  1. Lane 385: “LW/BW ratio between TAA-treated JnkΔhepa and JnkΔhepa animals” – one group should be Jnk fl/fl.

Reply: We apologize for overseeing this. We have now replaced the first Jnk∆hepa by Jnkf/f.

  1. Lanes 437 & 443: should it be indeed “Suppl. Fig. 8A”, or “Suppl. Fig. 9A”?

Reply: We apologize. As the Reviewer mentioned earlier both Suppl. Fig. 8A and 9A are the same. However, Suppl. Fig. 8A is a representation of the DEN/CCl4 model, whilst, Suppl. Fig. 9A refers to the treatment with CM272. This has now been corrected both in the text, and in the Suppl. Material section.

  1. Lane 447: instead of “(Suppl. Fig. 9A and B)” it seems, there should be “(Suppl. Fig. 9B and C)”

Reply: The Reviewer is right. It is ¨Suppl. Fig. 9B and C¨, now amended in the revised version of the manuscript.

REFERENCES

  1. Cubero, F.J.; Mohamed, M.R.; Woitok, M.M.; Zhao, G.; Hatting, M.; Nevzorova, Y.A.; Chen, C.; Haybaeck, J.; de Bruin, A.; Avila, M.A., et al. Loss of c-jun n-terminal kinase 1 and 2 function in liver epithelial cells triggers biliary hyperproliferation resembling cholangiocarcinoma. Hepatol Commun 2020, 4, 834-851.
  2. Colyn, L.; Barcena-Varela, M.; Alvarez-Sola, G.; Latasa, M.U.; Uriarte, I.; Santamaria, E.; Herranz, J.M.; Santos-Laso, A.; Arechederra, M.; Ruiz de Gauna, M., et al. Dual targeting of g9a and DNA methyltransferase-1 for the treatment of experimental cholangiocarcinoma. Hepatology 2021, 73, 2380-2396.

Reviewer 2 Report

In study links ER stress and fibrosis with cholangiocellular injury and cell death in JnkΔhepa mice, which are sensitive to CCA development. Therefore, the JnkΔhepa model can be a relevant experimental tool for the study of fibropolycystic liver diseases including CS. This work also identifies potential therapeutic strategies for a group of diseases lacking effective medical treatments.
I have some comments to this manuscript.
1.  
I understand that JNK deficiency seems to be important for the development of multiple liver cysts. Is there any evidence that JNK is important in clinical PLD?
2. There is no explanation for figure 1E.
3. 
How many weeks will liver cysts be observed in this model mouse from the start of observation? Show data whether there is a correlation between cysts and liver fibrosis.
4. 
Which of the UPR family, BIP, CHOP, sXBP1 or uXBP1 is more important? It seems that BIP is high in TM stimulation and CHOP is increased in TAA.
5. 
In Figure 8, cancer tissue was confirmed for the first time in the orthotopic transplant model. Is a tumor observed in the liver after administration of DEN / CCL4 to JNK-deficient mice? How does it make a difference compared to DEN / CCL4 administration to wild-type mice?
6. 
In Figure 9C, the fibrosis of the vehicle group is not strong, but how many weeks is the evaluation? Is CM272 effective against fibrosis caused by TM or TAA?

Author Response

POINT-TO-POINT RESPONSE TO REVIEWERS

Reviewer #2

In study links ER stress and fibrosis with cholangiocellular injury and cell death in JnkΔhepa mice, which are sensitive to CCA development. Therefore, the JnkΔhepa model can be a relevant experimental tool for the study of fibropolycystic liver diseases including CS. This work also identifies potential therapeutic strategies for a group of diseases lacking effective medical treatments.
I have some comments to this manuscript.

  1.  I understand that JNK deficiency seems to be important for the development of multiple liver cysts. Is there any evidence that JNK is important in clinical PLD?

Reply: We thank the Reviewer for this question. Our current data and previous publications [1] undoubtedly indicate that the JNK1/2 signaling in conjunction with other cell death effectors (RIPK1/CC8)–are pivotal in biliary regeneration and cellular plasticity in the liver and could enable the development of novel therapeutic strategies in polycystic liver diseases (PLD). Interestingly, the group of Dr. Luedde [1] detected RIPK1 expression in biliary epithelial cells of cystically dilated bile ducts of patients with Caroli disease/syndrome (with additional congenital hepatic fibrosis), suggesting that the regulation of JNK1/RIPK1 signaling could be a potential therapeutic avenue for PLD patients. We have added this information to the Discussion section of the revised version of the manuscript.

  1. There is no explanation for figure 1E.

Reply: We apologize for this mistake. We have corrected the text in the Figure legend. Fig. 1E refers quantification of positive cells from IF microphotographs of CD45 (left panel), F4/80 (center panel) and CD11b (right panel) in 8 to 72 week-old Jnkf/f and JnkΔhepa mice.

  1. How many weeks will liver cysts be observed in this model mouse from the start of observation? Show data whether there is a correlation between cysts and liver fibrosis.

Reply: We previously examined the progression of liver disease in hepatocyte-specific Jnk1/2 knockout mice (JnkΔhepa). Our results show that approximately 33% of these mice from week 30 of age histologically displayed cystogenesis in their liver parenchyma which did not affect liver function [2]. The presence of cysts in 52 weeks-old JnkΔhepa livers was found in 100% of mice. In the present study we show that, in parallel to cystogenesis, deposition of collagen fibers in the liver parenchyma of JnkΔhepa mice showed a clear tendency from 32 weeks of age and significant differences were found at 52 and 72 weeks of age, compared to Jnkf/f mice (Figure 1B). In addition, alpha-smooth muscle actin (αSMA) protein overexpression is evident at 32 weeks of age in Jnk∆hepa mice (Figure 1C) and increased mRNA expression levels of αSma, ColIa1 and Mmp9 were observed in ageing Jnk∆hepa compared with Jnkf/f livers in most of the time-points assessed (Figure 1D).  

  1. Which of the UPR family, BIP, CHOP, sXBP1 or uXBP1 is more important? It seems that BIP is high in TM stimulation and CHOP is increased in TAA.

Reply: This is a great point. As the Reviewer highlights BiP/GRP78, CHOP and XBP1 seem to be important in the cystogenesis and carcinogenesis in Jnk∆hepa mice (Figs. 4A-B, 5A). We focused on the IRE1α/XBP1/JNK pathway in this study. However, as the Reviewer mentions, more studies on the role of BiP and CHOP in the Jnk∆hepa mice phenotype are needed. In fact, we are currently planning and carrying out additional studies to further our understanding of the impact of other branches of the ER stress in PLD and carcinogenesis.

  1. In Figure 8, cancer tissue was confirmed for the first time in the orthotopic transplant model. Is a tumor observed in the liver after administration of DEN / CCL4 to JNK-deficient mice? How does it make a difference compared to DEN / CCL4 administration to wild-type mice?

Reply: This is an insightful comment, also made by Reviewer 1. As we have earlier shown [2], treatment of wildtype Jnkf/f mice with DEN/CCl4 triggers features of hepatocellular carcinoma (HCC) development. However, Jnk∆hepa mice treated with DEN/CCl4 develop cholangiocarcinoma (CCA) instead [3]. Thus, the main goal of this transplantation experiment was to fully evaluate the malignancy of these lesions and whether Jnk∆hepa – liver tissues were capable of recapitulating CCA histological characteristics in nude mice. Our results indicate that the engrafted implants displayed histological features resembling malignant CCA, as evaluated by an expert pathologist, Dr. August Vidal.

  1. In Figure 9C, the fibrosis of the vehicle group is not strong, but how many weeks is the evaluation? Is CM272 effective against fibrosis caused by TM or TAA?

Reply: The Reviewer is right in this appreciation. As mentioned in the text, these mice were sacrificed at 22 weeks. At this time, fibrosis in Jnk∆hepa mice is detectable but not considerable. Jnk∆hepa have significant fibrosis from 32 weeks. Regarding the protection of CM272 against TM and TAA these are great suggestions that we are willing to consider in a future experimental approach. Nevertheless, we have recently shown in other mouse models of liver fibrosis that CM272 consistently prevents the progression of liver fibrosis [4]. In the present study we provide evidence suggesting that CM272 can also have antifibrogenic activity in a model of fibrosis-related carcinogenesis.

REFERENCES

  1. Muller, K.; Honcharova-Biletska, H.; Koppe, C.; Egger, M.; Chan, L.K.; Schneider, A.T.; Kusgens, L.; Bohm, F.; Boege, Y.; Healy, M.E., et al. Jnk signaling prevents biliary cyst formation through a caspase-8-dependent function of ripk1 during aging. Proc Natl Acad Sci U S A 2021, 118.
  2. Cubero, F.J.; Mohamed, M.R.; Woitok, M.M.; Zhao, G.; Hatting, M.; Nevzorova, Y.A.; Chen, C.; Haybaeck, J.; de Bruin, A.; Avila, M.A., et al. Loss of c-jun n-terminal kinase 1 and 2 function in liver epithelial cells triggers biliary hyperproliferation resembling cholangiocarcinoma. Hepatol Commun 2020, 4, 834-851.
  3. Colyn, L.; Barcena-Varela, M.; Alvarez-Sola, G.; Latasa, M.U.; Uriarte, I.; Santamaria, E.; Herranz, J.M.; Santos-Laso, A.; Arechederra, M.; Ruiz de Gauna, M., et al. Dual targeting of g9a and DNA methyltransferase-1 for the treatment of experimental cholangiocarcinoma. Hepatology 2021, 73, 2380-2396.
  4. Barcena-Varela, M.; Paish, H.; Alvarez, L.; Uriarte, I.; Latasa, M.U.; Santamaria, E.; Recalde, M.; Garate, M.; Claveria, A.; Colyn, L., et al. Epigenetic mechanisms and metabolic reprogramming in fibrogenesis: Dual targeting of g9a and dnmt1 for the inhibition of liver fibrosis. Gut 2021, 70, 388-400.

Round 2

Reviewer 2 Report

There is a solid response to the reviewer's comments. The discussion is also more fulfilling than before. This revised version seems to be fine.